# Novel Methodology for Condition Monitoring of Gear Wear Using Supervised Learning and Infrared Thermography

**Emmanuel Resendiz-Ochoa, Juan J. Saucedo-Dorantes**, **Juan P. Benitez-Rangel,**
**Roque A. Osornio-Rios** and **Luis A. Morales-Hernandez ***

Faculty of Engineering, University Autonomous of Queretaro, San Juan del Rio 76807, Qro., Mexico;
eresendiz@hspdigital.org (E.R.-O.); jsaucedo@hspdigital.org (J.J.S.-D.); benitez@uaq.mx (J.P.B.-R.);
raosornio@hspdigital.org (R.A.O.-R.)
* Correspondence: lamorales@hspdigital.org

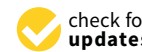

**Featured Application:** The application of the article lies in the monitoring and diagnosis of failures in gearbox and rotating machines.

**Abstract:** In gearboxes, the occurrence of unexpected failures such as wear in the gears may occur, causing unwanted downtime with significant financial losses and human efforts. Nowadays, noninvasive sensing represents a suitable tool for carrying out the condition monitoring and fault assessment of industrial equipment in continuous operating conditions. Infrared thermography has the characteristic of being installed outside the machinery or the industrial process under assessment. Also, the amount of information that sensors can provide has become a challenge for data processing. Additionally, with the development of condition monitoring strategies based on supervised learning and artificial intelligence, the processing of signals with significant improvements during the classification of information has been facilitated. Thus, this paper proposes a novel noninvasive methodology for the diagnosis and classification of different levels of uniform wear in gears through thermal analysis with infrared imaging. The novelty of the proposed method includes the calculation of statistical time-domain features from infrared imaging, the consideration of a dimensionality reduction stage by means of Linear Discriminant Analysis, and automatic fault diagnosis performed by an artificial neural network. The proposed method is evaluated under an experimental laboratory data set, which is composed of the following conditions: healthy, and three severity degrees of uniform wear in gears, namely, 25%, 50%, and 75% of uniform wear. Finally, the obtained results are compared with classical condition monitoring approaches based on vibration analysis.

**Keywords:** infrared thermography; thermal analysis; supervised learning; gearbox; sensor

## 1. Introduction

Sensors are important elements in automation and the new concept of industry 4.0 because they allow the monitoring and assessment of the physical conditions of equipment, in order to achieve a better capacity for the control, reliability and integrity of industrial equipment [1]. In recent years, sensing has become more important in the research field, resulting from technological advances in detection that allow multivariate sensor monitoring [2], generating significant improvement in the observability of industrial systems. However, multivariate monitoring in engineering systems provides a large amount of varied data, generating a challenge for data processing [3]. With the significant

development of intelligent detection techniques and artificial intelligence, data analysis offers a promising approach to effectively learning complex multivariable data, which is why supervised and unsupervised learning techniques may be considered as two powerful tools for solving problems presented as a result of a large amount of data to be processed, allowing the reduction of information and a significant improvement in the analysis and the classification of data [4].

Thus, condition monitoring and fault assessment strategies play an important role in ensuring the availability of industrial processes; in this regard, it is desirable that most industrial systems will be monitored by different sensors for ensuring the integrity of the industrial equipment. The gearbox transmission system is one of the important elements due to the fact that it contains mechanical elements that are commonly included in a wide variety of industrial applications; the reason for its common use is because it is needed in order to handle different rotating speeds and maintain specific torques between the electric machines and the load mechanics. Although these components are robust, efficient, reliable and have a low cost, their monitoring and continuous assessment are necessary [5]. Even so, the occurrence of unexpected faults in gearboxes can occur at any time, causing production losses, system malfunctions or unwanted downtime, with significant financial losses and human efforts [6].

Various faults may occur within the gearbox transmission system. In the literature, it has been reported that the occurrence of faults in the gears represents 80% in transmission machinery systems and 10% in rotating machinery systems [7], resulting from several factors ranging from inadequate lubrication, fluctuating loads, decoupling, misalignment, poor cooling, and gear design, among others [8]. In the literature, different methodologies for monitoring and detecting gearbox faults have been reported based on different sensing techniques and methods for analyzing and classifying data. For example, acoustic emissions are used to detect angular misalignments of shafts in the gearbox [9]. Furthermore, the influence of the oil film thickness has been considered in the detection of failures in helical gearboxes [10], and also the application of the discrete wavelet transform and the use of artificial neural networks for acoustic signals have been taken into account for the detection of faults in gearboxes [11]. On the other hand, vibration-based analysis represents the most classical approach to perform fault detection in gearbox transmission systems; in this sense, a time-frequency signal analysis using a generalized synchronism transformation is applied to diagnose gearbox failures and bearing defects from vibration signals [12,13], and intelligent diagnostic models for gearboxes have also been proposed from wavelet support vector machines and genetic algorithms using vibration signals [14]. Other physical magnitudes have been also considered; thus, the use of motor current signature analysis (MCSA (Table 1 shows the definition)) has also been a classical approach to identify the characteristic frequencies of gearbox failures [15] and also to diagnose eccentricity failures during transient speeds [16]. Yet, although some investigations have focused on the analysis of faulty conditions in gearbox transmission systems, there exists a lack of analysis to diagnose the occurrence of incipient faults, such as uniform wear in the gears, in such systems.

**Table 1.** Acronyms and definitions.

| Acronym | Definition |
| --- | --- |
| AC | Alternative Current |
| AI | Artificial Intelligence |
| ANN | Artificial Neural Network |
| DC | Direct Current |
| HTL | Healthy |
| k-NN | k-Nearest Neighbor |
| LDA | Linear Discriminant Analysis |
| LTSA | Load Torque Signature Analysis |
| MCSA | Motor Current Signature Analysis |
| MLP | Multilayer Perceptron |
| RTD | Resistance Temperature Detector |
| SVM | Support Vector Machine |

Gears are the main part of the gearbox transmission system because they are responsible for transmitting the circular motion through the contact of cogwheels inside of them. In spite of this, their continuous operation makes the occurrence of incipient failures inherent in this type of element. Indeed, the friction between the gears leads to reduction in the mechanical properties and accelerates the degradation of the gears; the appearance of uniform wear in gears is known as the most common and initial fault condition [17]. On the other hand, the most common faults that have been reported are related to the presence of irregularities in the teeth of the gears such as tooth breakage; chipping and cracks in the root; and chipping, pitting and damage to the surface of the tooth [18,19]. To detect these problems, adaptive wavelet filters are effectively applied to vibration signals acquired from the monitoring of the gearbox [20]. In addition, diagnosis methodologies based on vibration signals are proposed for the identification of wear in gears [7], with MCSA and load torque signature analysis (LTSA) being applied to detect faults such as misalignment, wear and mass imbalances of gears [21]. Nevertheless, few reported papers analyze and study gradual and uniform wear in the gears of the transmission system. Moreover, most techniques used for monitoring the condition in gearboxes have their limitations, with the vibration and the acoustic signal being affected by environmental noise and the location of the sensors [22]. MCSA is not a direct measurement of the equipment; it is based on the current of the electric motor, which can cause confusion when it comes to condition monitoring and failure detection in a gearbox [23]. Also, when a gear presents a fault, it manifests with an increase in temperature. One technique widely used for temperature monitoring is infrared thermography, and this may be a complementary and helpful method in conjunction with the most used and proven techniques for monitoring and detecting faults in a gearbox transmission system.

Thereby, infrared imaging is a noninvasive and nondestructive technique that efficiently monitors temperature and possesses a wide range of monitoring and the possibility to visualize and locate hot spots [24] through the increase in temperature caused by the faults present in the gearbox transmission system. Although this technology was expensive in the beginning, low-cost cameras and cores have emerged that make it more accessible and used in various areas such as medicine, manufacturing, and electrical engineering, among others [25–27].

Besides that, there are multivariable and multiple sensors, in which each pixel of the image represents a temperature value as well as a color intensity value. In the literature, few papers are reported that use infrared imaging for monitoring and diagnosis of gearbox failures. A thermal analysis was proposed based on MLP neural networks for the diagnosis of failures in helical gears [28]. Further, there was a review published discussing different studies with infrared imaging to detect failures in gearboxes [29], and infrared imaging was used as a complement to other techniques such as vibration signals and acoustic signals for the diagnosis of failures in a gearbox [30]. Even so, the papers reported in the literature that use infrared imaging only focus on failures in the gearbox and do not study the wear in the gears of the gearbox transmission system. Furthermore, not all the information generated by infrared imaging is used, and the analysis and classification of data may be chaotic. In addition, the use of artificial intelligence (AI) techniques has increased as an emerging field in industrial applications; in this regard, techniques such as k-Nearest Neighbor (k-NN), Naive Bayes Classifier, Support Vector Machine (SVM) and Deep Learning are used to represent an effective solution for the recognition and automatic diagnosis of failures in rotary machines. However, although most of them may be included in condition monitoring schemes, sometimes their improper application can produce several limitations; for example, k-NN needs a lot of storage space, Naive Bayes must have combinatorial and calculation problems and there is a need for prior probability, SVM has low efficiency, and Deep Learning needs large samples and training for a long time, plus it only focuses on failures in the gearbox. The proposal and choice of an AI technique such as an Artificial Neural Network (ANN) may lead to the performance of automatic fault diagnosis and condition assessment of the incipient occurrence of wear in gears, providing a high classification performance [31].

For this reason, the main contribution of this paper lies in the proposal of a novel noninvasive methodology that combines the analysis of thermographic images, supervised learning and artificial

neural network (ANN) for the automatic diagnosis and assessment of different levels of uniform wear conditions in a gearbox transmission system. Also, it helps to contribute and complement all the techniques used for the continuous diagnosis and identification of incipient faults in industrial machinery composed of gearboxes. Additionally, the novelty of this work includes the use of thermal images that come from a thermographic camera, which is used for the analysis of the thermal behavior of the gearbox. The thermal matrix of the thermography is used to extract a set of suitable features and to perform their dimensionality reduction through LDA to obtain a visual representation of the considered conditions; in addition, the implementation of an ANN allows the automatic classification and final diagnosis outcome. Thereby, the proposed methodology consists in detecting the hot spots of the infrared imaging produced by the uniform wear in the gears, and then extracting thermal statistical features that characterize the different levels of wear in the gears. Based on the extracted data sets, linear discriminant analysis (LDA) is applied for the reduction of dimensionality features. An ANN is proposed for the classification and detection of uniform wear in the gears. A novel methodology is applied to the study of four states in the gears: healthy (HTL), and with 25%, 50%, and 75% uniform wear in the gear. Finally, the novel methodology is compared to vibration signals, obtaining an improvement for the diagnosis and classification of faults by uniform wear in the gears of the gearbox transmission system when using infrared imaging.

## 2. Theoretical Considerations

### 2.1. Infrared Imaging

Infrared imaging is a technique used to record infrared radiation, which permits the estimation of a body's surface temperature. Any object at a temperature above absolute zero (0 K or −273.15 °C) emits energy electromagnetic radiation depending on its temperature [32]. An infrared imaging camera absorbs the infrared radiation emitted by a body through a noncontact method and, using Stefan–Boltzmann law, the body's temperature is obtained [33].

Additionally, the consideration of infrared cameras allows us to obtain either digitized intensity values or temperature values, and depending on the application, one or both values are used. In this sense, it should be mentioned that although several research studies exist that consider these values for carrying out condition monitoring and fault identification in rotating machinery, most of the reported works have not based the fault assessment on the estimation of the thermal matrix.

### 2.2. Linear Discriminant Analysis (LDA)

In regard to the condition monitoring of gearbox transmission systems, the estimation of a characteristic and significative set of features plays an important role in determining the occurrence of faults. Yet, although a high-dimensional set of features may be estimated to assess the actual condition of transmission systems, the calculation of non-useful and correlated information is inevitable. Thus, dimensionality reduction techniques play an important role in reducing high-dimensional sets of features and also for removing such non-significative and correlated information that may lead to low performances during the condition assessment [34]. Since the LDA technique is a supervised technique, its consideration in condition monitoring schemes is suitable due to the fact that it can face multi-class problems; moreover, through LDA an original *n*-dimensional feature space is reduced, aiming to maximize as much as possible the linear separation between the considered classes.

Indeed, such dimensional reduction is performed by means of a linear transformation, where the resulting low-dimensional space represents a linear combination containing different weights from the original features. Thus, to guarantee the maximum class separability, the ratio of the between-class

variance to the within-class variance is estimated [35]; thereby, by considering a multi-class problem with $C$ classes of $N$ number of samples, the between-class scatter matrix is calculated as follows:

$$S_b = \sum_{j=1}^{C} N_j \left( m_j - \overline{m} \right) \left( m_j - \overline{m} \right)^T \tag{1}$$

where $N_j$ belongs to the total number of samples for the $j$-th class $C_j$; considering all the evaluated classes, $\overline{m}$ is the global mean of all data samples, and $m_j$ is the local mean of each class $C_j$. Thus, the within-class scatter matrix is calculated as:

$$S_w = \sum_{j=1}^{C} \sum_{i=1}^{N_j} \left( x_i^j - m_j \right) \left( x_i^j - m_j \right)^T = \sum_{j=1}^{C} S_{w_j} \tag{2}$$

where $x_i^j$ is the $i$-th sample that corresponds to each class $C_j$; as a result, in $S_{wj}$, the corresponding covariance matrix of class $C_j$ is estimated.

Accordingly, the optimum and resulting vector of projection $W_{LDA}$ chosen by the LDA allows us to perform a good separation of the evaluated classes, since the estimated transformation matrix has orthonormal columns that maximize the ratio of the determinant of the between-class matrix of the projected samples to the determinant of the within-class scatter matrix of the projected samples, that is:

$$W_{LDA} = \arg \max \frac{\left| W^T S_b W \right|}{\left| W^T S_w W \right|} = \begin{bmatrix} w_1 & w_2 & \cdots & w_m \end{bmatrix} \tag{3}$$

where $\{ w_i | i = 1, 2, \cdots, m \}$ is the set of generalized eigenvectors, also known as discriminant vectors, of $S_b$ and $S_w$ that correspond to the $C$-1 largest generalized eigenvalues $\{ \lambda_i | i = 1, 2, \cdots, m \}$.

Thereby, the resulting extracted features represented in $V$ are calculated by means of projecting the original data set of features $X$ into the low-dimensional space $W_{LDA}$ as follows:

$$V = W_{LDA}{}^T X \tag{4}$$

### 2.3. Artificial Neural Network (ANN)

One of the popular artificial neural networks for pattern classification is the multilayered perceptron neural network (MLP). To define an ANN, it is necessary to establish parameters such as the connections, the number of layers, the activation functions, the propagation rules, etc. [32]. In the case of the MLP, it needs to consider its two different stages: the learning stage and the prediction process. In the case of the MLP, the propagation rule is the weighted sum, and it is defined according to (5).

$$\sum_{i=1}^{n} w_{ij} x_i(t) \tag{5}$$

where $w_{ij}$ is the weight that connects neuron $i$ in the input layer with neuron $j$ in the hidden layer, $x_i$ is the output from neuron $i$ in the input layer, $n$ is the number of neurons in the input layers, and $t$ is the pattern [36].

## 3. Methodology

In this section, we describe the proposed diagnosis methodology applied to the identification and classification of different levels of uniform wear in gears as an incipient failure in a gearbox transmission system. The methodology is mainly composed of six steps, as shown in Figure 1. In the first step, different conditions of uniform wear in gears are experimentally evaluated under continuous working conditions in a gearbox transmission system. Then, the second step considers

the continuous monitoring of the different conditions under evaluation, and from the continuous monitoring, infrared images are acquired, which contain meaningful information related to the current condition under assessment. Afterward, in the third step, the detection of hot spots over the infrared image is identified, aiming to compute its corresponding thermal matrix. Subsequently, in the fourth step, for each evaluated condition, a significative statistical set of features is estimated from the previous thermal matrices. Then, in the fifth step, a dimensionality reduction is carried out by means of linear discriminant analysis, aiming to obtain a visual representation of all considered conditions in a 2D space. Additionally, with the consideration of the dimensionality reduction stage, the classification task is facilitated for the considered classification algorithm. Finally, in the last step, automatic fault identification is performed by means of an artificial neural network; thus, the condition assessment results in the classification of the different levels of uniform wear in the gears of the gearbox transmission system.

### 3.1. Condition Monitoring

During the continuous condition monitoring, different levels of uniform wear are iteratively evaluated in the gearbox transmission system; in this regard, several hot spots are identified over the gearbox transmission system in order to obtain the most significative information related to its thermal behavior for each evaluated condition. Once the identification of the hot spots has been performed, the system is prepared to be monitored.

### 3.2. Data Acquisition

A thermographic infrared camera (FLIR A310 [9 Hz], FLIR Systems Inc., Wilsonville, Oregon, USA) is used as the primary sensor for data acquisition; therefore, the continuous monitoring of the thermal behavior of the gearbox transmission system under evaluation is performed by capturing images. In this regard, as a result of the acquisition and capture of images, two different images are obtained: first, a pseudo-color infrared image related to the digitized intensity values, and second, a gray-scale infrared image, which is associated with the temperature values. Thus, for the proposed condition monitoring strategy, gray-scale infrared images are used to estimate the corresponding thermal matrix.

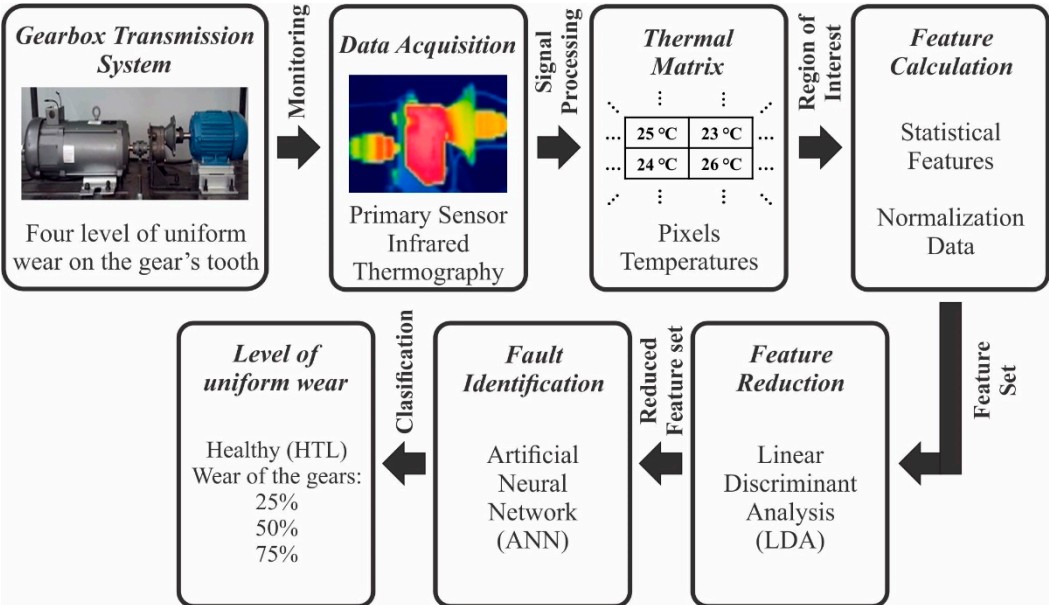

**Figure 1.** Methodology proposed to identify gear tooth wear in a gearbox transmission system.

### 3.3. Thermal Matrix

As aforementioned, the consideration of infrared cameras allows the obtainment of either digitized intensity values or temperature values; in this sense, it should be mentioned that in this proposed work, gray-scale infrared images are used to estimate the thermal matrix due to its simplicity. Therefore, the procedure of calculation of the thermal matrices is based on the estimation of the true temperature (thermogram) from each pixel that composes the gray-scale infrared image (Figure 2).

$$T_{true}(x,y) = Tc_{min} + \frac{T_{gray}(x,y)}{T_{mgv}} * (Tc_{max} - Tc_{min}) \tag{6}$$

where $T_{true}(x,y)$ is the value of the true temperature derived from the pixel intensity, $Tc_{max}$ and $Tc_{min}$ are the maximum and the minimum temperature of the infrared image, respectively, $T_{gray}(x,y)$ is the value of the pixel intensity in the gray-scale image, and $T_{mgv}$ is the peak intensity value in the infrared image.

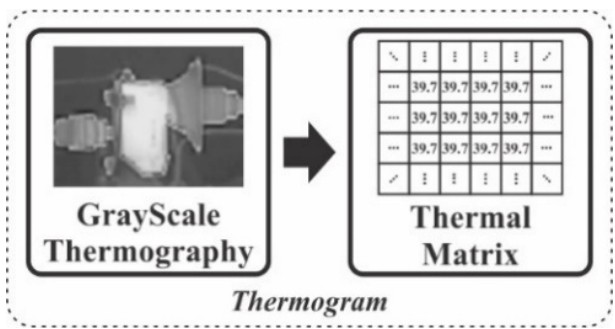

**Figure 2.** Thermal matrix of the infrared thermography.

### 3.4. Calculation of Features

Commonly, condition monitoring strategies are based on the calculation of a high-dimensional set of features from different acquired physical magnitudes [37]. However, in this proposed work, we consider the estimation of a specific set of significant features to characterize the thermal behavior of the gearbox transmission system under assessment; thus, the set of features is: maximum temperature, minimum temperature, and average temperature; and the analysis of the asymmetry of thermographic images is also considered based on the calculation of characteristics' stated features of first statistical order (average histogram, medium, standard deviation, variance, skewness, kurtosis, entropy, and energy). In addition, these features have the ability to adapt to changes according to the operating conditions of the system.

Table 2 summarizes the mathematical equations of the set of considered features to characterize the thermal behavior of the gearbox transmission system. In these equations, $f_x$ indicates the number of pixels per column (width of the image), and $g_y$ the number of pixels per row (height of the image); $p$ is the number of distinct gray levels in the quantized image, $h(p)$ is the intensity of the pixel in the histogram, and $q$ is the level of intensity of the image.

On the other hand, the average intensity determines the brightness or darkness of the image, skewness defines the degree of asymmetrical property of the histogram with respect to average intensity, kurtosis measures the peakness or flatness of the intensity distribution with respect to the normal distribution, entropy measures the randomness of the input image, and variance outlines the deviation of gray-level pixels from the mean [32].

**Table 2.** Extraction of statistical features from the thermography image.

| Feature | Equation | No. |
|---|---|---|
| Maximum value | $T_{max} = \max(I(x,y))$ | (7) |
| Minimum value | $T_{min} = \min(I(x,y))$ | (8) |
| Average intensity | $\mu = \frac{1}{f_x g_y} \sum\limits_{p=0}^{q-1} ph(p)$ | (9) |
| Standard deviation | $\sigma = \left( \frac{1}{f_x g_y} \sum\limits_{p=0}^{q-1} (p-\mu)^2 h(p) \right)^{\frac{1}{2}}$ | (10) |
| Variance | $\sigma^2 = \frac{1}{f_x g_y} \sum\limits_{p=0}^{q-1} (p-\mu)^2 h(p)$ | (11) |
| Skewness | $S_k = \frac{1}{\sigma^3 f_x g_y} \sum\limits_{p=0}^{q-1} (p-\mu)^3 h(p)$ | (12) |
| Kurtosis | $k = \frac{1}{\sigma^4 f_x g_y} \sum\limits_{p=0}^{q-1} (p-\mu)^4 h(p)$ | (13) |
| Entropy | $Entropy = -\frac{1}{f_x g_y} \sum\limits_{p=0}^{q-1} n \, log(p)$ | (14) |
| Energy | $Energy = \frac{\sum_{o=0}^{f_x} \sum_{q=0}^{g_y} I(x,y)^2}{f_x g_y}$ | (15) |

### 3.5. Reduction of Features

After carrying out the feature calculation, a dimensionality reduction stage, by means of LDA, is also included in the proposed condition monitoring methods. Such a dimensionality reduction, which is based on a linear transformation, allows us to obtain a visual representation of all the evaluated conditions in a 2D space. Therefore, the new extracted features in the 2D space retain the most significative and discriminative information of the original set of statistical features since the new extracted features are composed of a combination containing different weights from the original features. Thus, these non-useful or correlated features may have low weights, whereas the significative features will have greater weighting.

### 3.6. Fault Identification

Once the most significant and discriminant information has been highlighted by LDA, all the considered conditions are represented in a 2D space, where the linear separation between classes is retained as much as possible. After that, automatic fault identification is carried out by means of an Artificial Neural Network (ANN); as a result, the final diagnosis outcome results in the classification of four different conditions; HTL, and 25%, 50%, and 75% wear in the gears. Since the classification task is facilitated by considering the dimensionality reduction through LDA, the structure of the ANN classifier is based on a classical structure, which only considers the input layer, a single hidden layer, and the output layer.

## 4. Experimental Setup

In this paper, an experimental test bench was used to test different levels of uniform wear in the gears as an incipient failure in a gearbox transmission system. Figure 3 shows a kinematic chain with the principal elements that an industrial system can have, where the main element is the gearbox, which is used to validate the proposed methodology. The test bench is based on a kinematic chain that consists of a three-phase induction motor of 1.5 kW (WEG00236ET3E145T-W22), electrically connected to a frequency inverter (VFD) (WEGCFW08) to feed and control the rotation speed. The AC machine is mechanically coupled by rigid coupling to a 4:1 gearbox (BALDOR GCF4 × 01AA) that drives its input shaft, and this gearbox is used to test the different levels of uniform wear in the gears studied in this paper. Besides, the gearbox, in turn, is mechanically coupled by rigid coupling to a DC generator

motor (BALDOR CDP3604), and the generator is used as a load, producing approximately 20% of the rated load in the induction motor in working conditions. For the monitoring of the thermal behavior of the gearbox, a thermographic infrared camera (model FLIR A310, from FLIR Systems Incorporated) is used as the primary sensor for data acquisition. The FLIR A310 has a resolution of 320 × 240 (76,800) pixels, and by means of a vanadium oxide microbolometer, detects infrared radiation with a thermal sensitivity/Noice Equivalent Tempearture Difference (NETD) of 50 mK. The measurements carried out with this camera are obtained with an accuracy of ±2 °C, and it is capable of measuring the object's temperature within a range of −20 to +120 °C or 0 to +350 °C. The acquired thermal images represent the digitized intensity values in 16 bits, and the FLIR Tool software is used for acquiring the thermographic images. The infrared camera is adjusted for each test to different environmental factors for a more accurate measurement. These factors are emissivity, atmospheric temperature, relative humidity, reflected temperature, and the distance between the gearbox and the infrared camera. For each test performed, measuring instruments are used, such as Fluke 975 AIMETER and Fluke 61, for the measurement of environmental parameters, while the emissivity value is adjusted to 0.95, as recommended in previous works related to electrical systems [38]. Calibration and validation are performed for the thermographic temperature values of the camera. For the calibration, the temperature values obtained with the thermographic camera are compared with the output of a comparative method. Several Resistance Temperature Detectors (RTDs) were used to obtain the reference measurement. The calibration guarantees that the measurement obtained with the thermographic camera is the same as that of the equipment to be monitored. For this paper, a thermal image is captured every minute from the start to the end of the test, in order to monitor the thermal behavior of the gears of the gearbox transmission system.

In this paper, an alternative study of four different wear conditions is proposed in the gears of the gearbox transmission system: healthy (HLT), and 25%, 50%, and 75% uniform wear, respectively. To produce the condition of wear failure in the gears, a gear factory was commissioned to manufacture it artificially.

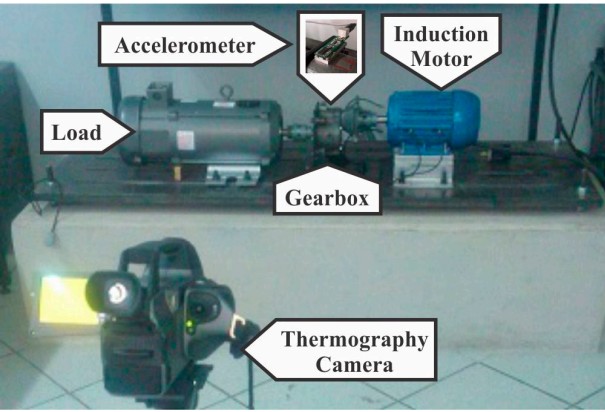

**Figure 3.** Experimental test bench used to test and recognize gearbox wear.

To produce the faults, the gears undergo a machining process where all their teeth are used by a tungsten cutter. Then, these gears are also subjected to a lapping process with the aim of making the wear induced on the gears as real as possible. Figure 4 shows the gears used for the tests carried out in this paper, with three different levels of uniform wear (25%, 50%, and 75%) and a healthy condition (HLT) being implemented to demonstrate the effectiveness of the proposed diagnostic methodology.

Four experimental conditions are studied on the gearbox: HLT, and 25%, 50%, and 75% wear. Each test lasted 90 min, since in this time, the healthy motor reached thermal stability.

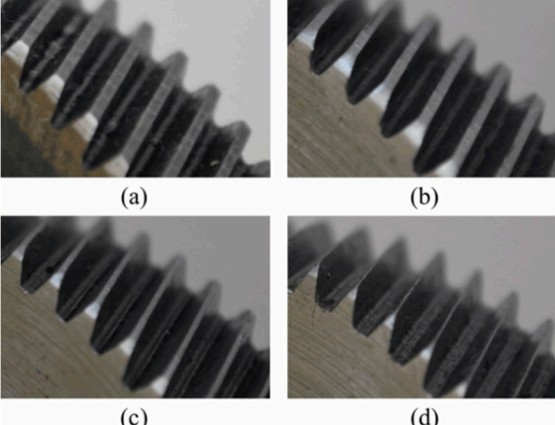

**Figure 4.** Proposed study cases for evaluating the gear in the gearbox: (**a**) HTL; (**b**) 25%, (**c**) 50%, and (**d**) 75% of uniform wear in the gears.

## 5. Results

The diagnosis methodology proposed for the assessment of different levels of uniform wear in a gearbox transmission system was implemented in Matlab (Matlab 2017a, MathWorks Inc.), which was used for the processing of thermographic images, the estimation of statistical features, the reduction of features with LDA, and the ANN integration to perform the fault classification.

As part of applying the proposed condition monitoring strategy, during the experimental evaluation of the different levels of uniform wear in gears, a set of infrared images was acquired for each case under study. Therefore, during the monitoring of each condition, each one of the tests was run for 90 min, acquiring a thermographic image every minute, and for each condition the last 50 acquired infrared images from the last 50 min were acquired and stored; therefore, a database of 200 samples (infrared imaging) was acquired. In this regard, it should be highlighted that only the last 50 infrared images were acquired due to the first 40 min being associated with the time that the gearbox transmission system requires to reach its thermal stability.

Subsequently, after the experimental evaluation of each considered condition, a complete database of the thermal behavior of the gearbox transmission system was acquired. Thus, Figure 5a–d shows the last thermal image acquired during the monitoring of each considered condition; these images belong to the last acquisition obtained at the 90th minute. In this regard, Figure 5a corresponds to the thermal monitoring of the gearbox when the gears are in a HTL state, Figure 5b corresponds to the condition of the gears with 25% uniform wear, Figure 5c represents the infrared imaging observed with 50% uniform wear of the gears, and Figure 5d shows the thermal behavior of the gearbox with 75% uniform wear of the gears. Additionally, from these Figures, it is possible to observe that there exist some differences that should be considered to provide the fault assessment; yet, although these thermal images show differences, an improved processing may lead to an accurate diagnosis.

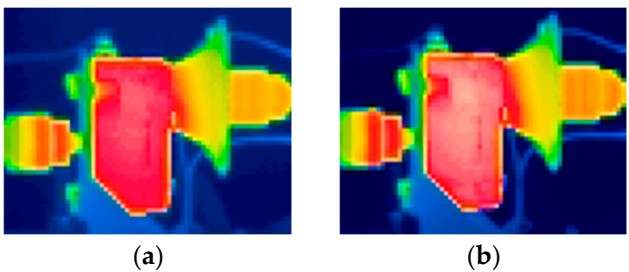

(**a**)　　　　　　　　　　　　　　　　　(**b**)

**Figure 5.** *Cont.*

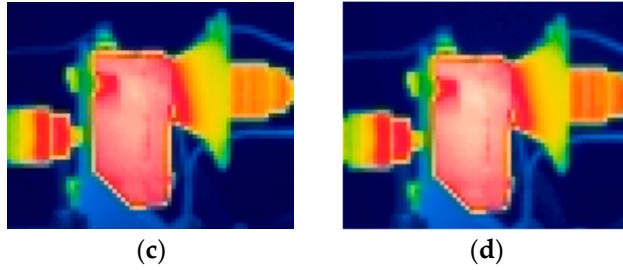

**Figure 5.** Acquired signals for the condition monitoring of gears in a gearbox transmission system with infrared thermography: (**a**) HTL gears, and (**b**) 25%, (**c**) 50%, and (**d**) 75% uniform wear in the gears.

Once the thermography images have been acquired, all the acquired images are then characterized by obtaining their thermal matrices. Then, from each thermal matrix, a set of statistical features is estimated, which for this proposed work are named as thermal statistical features. Therefore, each considered condition is now characterized by a consecutive set of thermal statistical samples. Despite the high level of characterization provided by the set of thermal statistical features, not all of them contain the same representative information associated with the occurrence of uniform wear in the gears of the gearbox transmission system under assessment. In this sense, in order to preserve the best thermal statistical features extracted from the thermographic images, the estimated sets of statistical features are processed through a dimensionality reduction approach, which allows us to retain the most significant and discriminate information. In this regard, the LDA strategy is applied to the original set of statistical features estimated from the thermal matrices; as a result, a visual representation of all considered conditions in a 2D space is obtained.

Figure 6 shows the resulting projection of the new extracted set of features obtained by means of applying LDA onto the original data sets that characterize the uniform wear conditions in the gears of the gearbox transmission system, such as a healthy gear, and 25%, 50%, and 75% wear. From this Figure, it is possible to observe that all the considered conditions appear separated from each other. Moreover, it should be mentioned that this resulting 2D projection results from a linear combination containing different weights from the original features, where the most significative features have a greater weighting value.

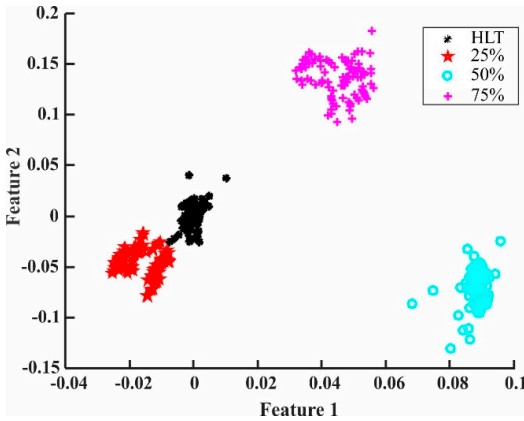

**Figure 6.** Projection of features, reduced by LDA, that characterize the condition of the gearbox onto the data set of infrared thermography.

After obtaining the data set reduced into 2D space with the LDA strategy, the automatic fault classification is carried out through the ANN-based algorithm. In this sense, due to the consideration of the feature reduction stage by means of LDA, which allows the obtainment of a visual representation of all considered conditions in a 2D space and facilitates the classification task, the structure of the considered ANN classifier is based on a classical structure composed only of the input layer, a single

hidden layer, and the output layer. The input layer consists of two neurons, the hidden layer has ten neurons as the classical implementation of ANN classifiers suggests, and the output layer is composed of four neurons.

Subsequently, in order to obtain statistically significant results, the ANN-based classifier is trained and validated under a five-fold cross-validation scheme. In this regard, taking into consideration all the conditions evaluated (HTL, and 25%, 50% and 75% uniform wear of gears), the original database that consists of 200 samples, 50 samples per condition, is divided into two different data sets for training and validation purposes. Therefore, the first data set used for training is composed of 160 samples, 40 samples per condition, while the data set used for validation consists of 40 samples, 10 samples per condition. To analyze the performance of the classification, a five-fold cross-validation scheme is applied to determine the variability of the training and the validation data of the classifier. Therefore, four types of classification are averaged, obtaining 100% in the classification index for both training and validation.

Tables 3 and 4 summarize the confusion matrices obtained during the training and validation of the proposed ANN classifier. As it can be noted in these Tables, the global classification ratio achieved during the training and validation is about 100%. Moreover, the consideration of the ANN as a classifier also allows the calculation of the classification regions, which provides a visual representation of the region that belongs to each evaluated condition. In this sense, Figure 7a,b shows the classification regions obtained by the ANN classifier during the training and validation procedures, respectively.

**Table 3.** Confusion matrix for the training of the ANN classifier with infrared imaging.

|  |  | Target Class | | | |
|---|---|---|---|---|---|
|  |  | HTL | 25% | 50% | 75% |
|  | HTL | 40 | 0 | 0 | 0 |
| Output Class | 25% | 0 | 40 | 0 | 0 |
|  | 50% | 0 | 0 | 40 | 0 |
|  | 75% | 0 | 0 | 0 | 40 |

**Table 4.** Confusion matrix for the validation of the ANN classifier with infrared imaging.

|  |  | Target Class | | | |
|---|---|---|---|---|---|
|  |  | HTL | 25% | 50% | 75% |
|  | HTL | 10 | 0 | 0 | 0 |
| Output Class | 25% | 0 | 10 | 0 | 0 |
|  | 50% | 0 | 0 | 10 | 0 |
|  | 75% | 0 | 0 | 0 | 10 |

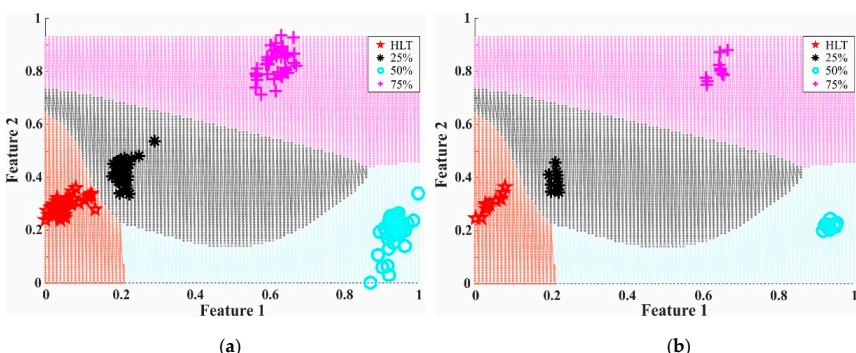

(a)    (b)

**Figure 7.** Resulting projection of the classification regions obtained during the identification of uniform wear in gearbox gears by means of the ANN-based classifier: (**a**) training procedure and (**b**) validation procedure.

## 6. Comparative Methodology

In order to highlight the contribution of the proposed condition monitoring based on the analysis of acquired infrared images, the obtained results have been compared with classical condition monitoring approaches that propose the estimation and evaluation of a high-dimensional set of features and that also propose the analysis of vibration signals. Therefore, regarding the consideration of a high-dimensional set of features, the set of thermal statistical features estimated from the thermal matrices is directly evaluated under the same structure of the ANN classifier proposed in this work. As a result, Tables 5 and 6 summarize the confusion matrices estimated by the proposed ANN structure when the original set of statistical thermal features, estimated from the thermal matrices, is directly assessed. From these obtained results, it should be highlighted that the exclusion of the dimensionality reduction stage, performed by LDA, significantly affects the classification performance, and although all the data variance of the original set of thermal features is evaluated by the ANN structure, a more complex structure of ANN classifier must be required to perform accurate results. Subsequently, during the training and evaluation of the original set of statistical thermal features, classification ratios of about 86.2% and 60% were achieved, respectively. Therefore, it is demonstrated that including a dimensionality reduction stage, by means of LDA, allows for an improved characterization of the considered conditions due to only the most significative and useful information being retained; additionally, the classification task is facilitated, resulting in the consideration of a classical structure for the proposed ANN structure. Indeed, when the original set of features estimated from thermal matrices is directly evaluated with the same structure of the proposed ANN classifier, a low classification performance is obtained due to the original set of features requiring the consideration of a complex structure of ANN classifier for modeling all the considered conditions.

**Table 5.** Confusion matrix for the training of the ANN classifier with extraction of statistical features.

|  |  | Target Class | | | |
|---|---|---|---|---|---|
|  |  | HTL | 25% | 50% | 75% |
|  | HTL | 33 | 0 | 10 | 0 |
| Output Class | 25% | 0 | 39 | 1 | 0 |
|  | 50% | 7 | 0 | 28 | 2 |
|  | 75% | 0 | 1 | 1 | 38 |

**Table 6.** Confusion matrix for the validation of the ANN classifier with extraction of statistical features.

|  |  | Target Class | | | |
|---|---|---|---|---|---|
|  |  | HTL | 25% | 50% | 75% |
|  | HTL | 6 | 0 | 4 | 8 |
| Output Class | 25% | 4 | 10 | 0 | 0 |
|  | 50% | 0 | 0 | 6 | 0 |
|  | 75% | 0 | 0 | 0 | 2 |

On the other hand, the results of the proposed condition monitoring methodology are also compared with classical monitoring approaches that consider the analysis of vibration signals; therefore, when the considered conditions were experimentally evaluated, at the same time that the infrared images were captured, mechanical vibration signals were also continuously acquired by means of a triaxial accelerometer model LIS3L02AS4. This accelerometer was installed on the top of the gearbox to perform continuous vibration acquisition in the perpendicular plane of the axis of rotation in the gearbox. The vibration signals were acquired at a sampling frequency of 3 kHz during 90 s of the continuous working condition of the gearbox transmission system.

Once the vibration signals had been obtained, the proposed condition monitoring methodology was applied by taking into account the acquired vibration signals instead of the thermal images;

therefore, from the acquired vibration signals, the proposed set of statistical features was also estimated, and then the feature reduction was carried out through LDA. Subsequently, all the considered conditions, represented by the characterization of vibration signals acquired during the assessment of gearbox condition, were represented in a 2D space.

Figure 8 shows the resulting projection of the estimated set of statistical features estimated by considering vibration signals and then reduction by LDA. From this obtained projection, it is possible to observe that an overlapping between the healthy condition and a faulty condition is obtained; also, there is an overlapping between two of the faulty conditions. These overlapping problems will be reflected in a low performance of classification. Subsequently, the same structure of the ANN classifier was used to carry out the automatic fault diagnosis, and the obtained classification matrices achieved during the training and validation are summarized in Tables 7 and 8, respectively.

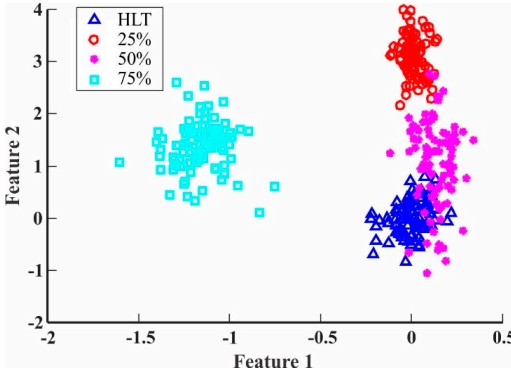

**Figure 8.** Projection of features, reduced by LDA, that characterize the condition of the gears onto the data set of vibration signals.

Moreover, as the proposed method considers the classification performance of the ANN classifier when vibration signals are also analyzed by considering a five-fold cross-validation scheme, the resulting classification ratios achieved during the training and validation are about 81.2% and 83.5%, respectively. Therefore, as was expected, the obtained results show a critical reduction in the classification performance when vibration signals are considered; in this regard, the consideration of thermal imaging makes the proposed methodology suitable to be applied in industrial environments, with a high performance of assessment and also with the advantage of non-interruption of the working condition due to the noninvasive monitoring.

**Table 7.** Confusion matrix for the training of the ANN classifier with vibration signals.

|              |      | Target Class |     |     |     |
| ------------ | ---- | ------------ | --- | --- | --- |
|              |      | HTL          | 25% | 50% | 75% |
|              | HTL  | 30           | 0   | 11  | 0   |
| Output Class | 25%  | 0            | 34  | 5   | 0   |
|              | 50%  | 10           | 6   | 24  | 0   |
|              | 75%  | 0            | 0   | 0   | 40  |

**Table 8.** Confusion matrix for the validation of the ANN classifier with vibration signals.

|              |      | Target Class |     |     |     |
| ------------ | ---- | ------------ | --- | --- | --- |
|              |      | HTL          | 25% | 50% | 75% |
|              | HTL  | 6            | 0   | 0   | 0   |
| Output Class | 25%  | 4            | 10  | 3   | 0   |
|              | 50%  | 0            | 0   | 7   | 0   |
|              | 75%  | 0            | 0   | 0   | 10  |

## 7. Discussion

As presented in the literature review, different condition monitoring strategies have focused on detecting the occurrence of faulty conditions in gearbox transmission systems; indeed, the occurrence of most of the addressed conditions is related to inadequate lubrication, eccentricity failures, fluctuating loads, decoupling, misalignment, poor cooling, and gear design, among others [8–11]. However, most of the reported methodologies are limited to the diagnosis of gearbox faults as a single gear tooth damage, whereas most analyzed conditions are related to the presence of irregularities in the teeth of the gears such as tooth breakage; chipping and cracks in the root; and chipping, pitting and damage to the surface of the tooth. [7,19–21]. In fact, from an industrial viewpoint, these faulty conditions may be considered as a critical condition in gearbox transmission systems since their occurrence may produce critical damage to the entire transmission system, leading to the machine breakdown. Therefore, as the literature review depicts, few works have focused on analysis for monitoring and assessing the appearance of incipient faults, such as uniform wear, in gearbox transmission systems, which is the main objective of this proposed work.

In this regard, [39] proposes a multidimensional hybrid intelligent method for gear fault diagnosis; this proposal includes the estimation of time-domain, frequency-domain and time-frequency-domain features, from acquired vibration signals, through the Hilbert transform, the wavelet packet transform (WPT) and the empirical mode decomposition (EMD); moreover, by means of multiple classifiers combined with a genetic algorithm (GA), different levels of damage are identified in the tooth root. Yet, although different severities of incipient damage in the tooth root are identified, the estimation of frequency-domain and time-frequency-domain features requires additional knowledge since their estimation is performed through complex signal processing techniques. In this sense, for the proposed work, the consideration of only statistical time-domain features leads to a high-performance characterization of the acquired thermal images.

On the other hand, a preview research work that assessed uniform wear in gears has also been performed by our research group [7]; in this study, the assessment of the uniform wear in gears was carried out by means of vibration signals. The novelty of this work includes the consideration of feature selection and feature reduction stages, and the final diagnosis outcome was performed by means of a fuzzy-based classifier, resulting in an average of 97.27% for the global classification ratio. Yet, although the detection of different levels of uniform wear condition is performed, the application of such a proposed method includes the installation of vibration sensors that represent a critical limitation in industrial environments. Thereby, in order to face the consideration of invasive sensors, such as accelerometers, the proposed condition monitoring presented in this work represents a suitable option that leads to the obtainment of accurate responses during the assessment of incipient faults in gearbox transmission systems.

Additionally, an improvement in the diagnosis and fault identification is also obtained in the proposed work by taking into account continuous monitoring through infrared images; thus, the obtained results make the proposed methodology appropriate to be considered as an attractive, alternative tool for classical infrared imaging inspection procedures and for techniques used for the diagnosis and classification of different levels of uniform wear in gears, such as an incipient failure in a gearbox transmission system.

## 8. Conclusions

This paper presents a novel methodology based on noninvasive monitoring for assessing different levels of uniform wear in a gearbox transmission system. There are four important point that must be highlighted in this new proposed methodology. The first is the reliability of using infrared images and the statistical features of hot spots, since they allow us to obtain a more suitable characterization of the operation of the gears in terms of thermal behavior, due to the fact that the temperature increase in the gears depends on the work performance of the uniform wear condition of the gears. The second is the implementation of dimensionality reduction strategies for the optimization of the set of statistical

features estimated from the infrared imaging, since it allows the elimination of less discriminant features and the compression of more significant statistical features. The third is the classification based on an artificial neural network, which is capable of identifying and classifying the different case studies proposed in this paper for the study of uniform wear in the gears of the gearbox transmission system. The fourth is the comparison that is made between infrared imaging and vibration signals to determine the reliability in the results for the diagnosis of failures, obtaining 100% of the total classification index when applying the novel methodology using infrared imaging, due to the good performance that was obtained both in the training and validation of the classifier. Meanwhile, for the novel methodology with vibration signals, a classification index of 81.2% was obtained.

The obtained results indicate that the proposed condition monitoring strategy is suitable to be applied in the assessment of gearbox transmission systems in industrial applications, where the use of invasive sensors represents a critical limitation. Further research will implement an online diagnosis methodology for assessing faults in gears and other elements of the gearbox transmission system.

**Author Contributions:** J.J.S.-D. conceived and designed the experiments; J.J.S.-D. and E.R.-O. performed the experiments; E.R.-O. analyzed the data; J.P.B.-R., R.A.O.-R. and L.A.M.-H. contributed reagents/materials/analysis tools; E.R.-O., J.J.S.-D., R.A.O.-R. and L.A.M.-H. wrote the paper. All authors have read and agreed to the published version of the manuscript.

**Funding:** This research was funded by [PRODEP] and CONACYT [434850].

**Conflicts of Interest:** The authors declare no conflict of interest.

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
