# Peer review of "Novel Methodology for Condition Monitoring of Gear Wear Using Supervised Learning and Infrared Thermography"

_applsci, doi:10.3390/app10020506_

Round 1

Reviewer 1 Report

The paper presents thermography investigations of gears with different damage degree. The thermography data are evaluated by artificial neural network to classify the damage degree.

The topic is interesting, but there are several points which should be clarified in the paper:

Lot of acronyms are used in the paper (LDA, LTSA, SVM, RTD,..) used in the paper, it would be useful to have a summary table of them. Line 107: “These techniques do not study the wear on the gears of the gearbox transmission system.”: the listed techniques, as SVM, ANN, k-NN, do not study any specific system,  they are general tools which can be used for classification. This sentence is confusing, please delete or correct it. Line 135: “An important point in the use of infrared imaging is that most of the reported papers use only the color intensity of the images and not the thermal matrix that is important.” : Infrared cameras deliver either digitized intensity values or temperature values, which one used, depends on the application and there are many papers handling with real temperature values. Whether the digitized intensity images or the thermal images are shown as a (pseudo) color image or a gray-scale image, has nothing to do with the content of the image. Please correct or delete this sentence. Line 175: “With the image capture, two additional images can be obtained: first, a pseudo color infrared imaging to process and second, a gray-scale infrared imaging which is used by the thermal matrix.”: What do you mean with ‘two additional images’, additional to what? See my previous comment: whether pseudo color or gray-scale image is displayed, has nothing to do, whether digitized intensity values or temperature values are shown. Please give more details to the camera: it is probably an uncooled microbolometer camera. Number of pixels? Temperature resolution (NETD)? Digitized value 8 or 14 bits? Used software? Line 242: “the emissivity value is adjusted to 0.93, recommended in previous works related to electrical systems [38]”: Ref .38 does not deal with electrical systems but thermal investigation of human body. The human skin has really a high emissivity value, but metal surfaces have usually much lower emissivity than 0.93. 5.: are these images after 90 min. at the end of the tests? LDA is an important part of the paper, it should be described in more detail. In Table 1 nine statistical features are listed, but how the authors get from these nine ‘feature 1’ and ‘feature 2’ in Fig.6, is not presented. Please add a more detailed descriptions, how the feature dimensionality reduction was carried out. Why are Fig.6 and Fig.7 different? As well the points as also the range of the features are different. Please explain more detailed the steps between. Line 307: 200 samples, 50 samples per condition were used. How these samples were got? Always at the end of the 90 min inspection time and 200 separate measurements, or selected time points from the 90 min? Please give a more precise description. ANN was for the training (l.304): how many layers and how many neurons were used in each layer? In Section 6 the statistical features were used without LDA for classification. What kind of ANN was used here (number of layers and neurons)? It is very strange that the performance of this ANN is so much worse. Due to LDA no more information is given to the trained system only the inputs are simplified. Therefore probably a more complex NN is necessary and more iteration to achieve the same results, but otherwise it is not understandable why the performance should be worse.

Reviewer 2 Report

The article presents a neural classifier, but the structure of the network used in the research is not provided (number of layers, number of neurons, graph of network learning error, etc.) The description of the used neural network should be completed.

Reviewer 3 Report

Authors did not compare their results with the results of other researchers. This is not the research paper, if the section "discussion" is omitted. Authors have to add the section: Discussion to the text and compary their results with results from different papers.

Otherwise the paper will be rejected.

Round 2

Reviewer 1 Report

All my comments were regarded and well answered.

In Fig.7a and 7b now the legend is missing.

Minor spell check is required, as e.g.

considerartion(l.419), dimentionality (l.419), taks (l.421), obatained (l.424) ,

consided (l.456), interupting(l.462)

Reviewer 3 Report

The discussion section is poor. The Authors still did not presented results from other research and still did not compare Their our results with other results.
